# Serous Membrane Detachment with Ultrasonic Homogenizer Improves Engraftment of Fetal Liver to Liver Surface in a Rat Model of Cirrhosis

**DOI:** 10.3390/ijms222111589

**Published:** 2021-10-27

**Authors:** Yumi Kawakatsu-Hatada, Soichiro Murata, Akihiro Mori, Kodai Kimura, Hideki Taniguchi

**Affiliations:** 1Department of Regenerative Medicine, Yokohama City University Graduate School of Medicine, 3-9 Fukuura, Kanazawa-ku, Yokohama 236-0004, Japan; y_hatada@yokohama-cu.ac.jp (Y.K.-H.); t176065b@yokohama-cu.ac.jp (A.M.); t186506f@yokohama-cu.ac.jp (K.K.); rtanigu@g.ecc.u-tokyo.ac.jp (H.T.); 2Division of Regenerative Medicine, Center for Stem Cell Biology and Regenerative Medicine, The Institute of Medical Science, The University of Tokyo, 4-6-1 Shirokanedai, Minato-ku, Tokyo 108-8639, Japan

**Keywords:** liver transplantation, liver cirrhosis, serous membrane, Cavitron Ultrasonic Surgical Aspirator

## Abstract

Liver transplantation is the most effective treatment for end-stage cirrhosis. However, due to serious donor shortages, new treatments to replace liver transplantation are sorely needed. Recent studies have focused on novel therapeutic methods using hepatocytes and induced pluripotent stem cells, we try hard to develop methods for transplanting these cells to the liver surface. In the present study, we evaluated several methods for their efficiency in the detachment of serous membrane covering the liver surface for transplantation to the liver surface. The liver surface of dipeptidyl peptidase IV (DPPIV)-deficient rats in a cirrhosis model was detached by various methods, and then fetal livers from DPPIV-positive rats were transplanted. We found that the engraftment rate and area as well as the liver function were improved in rats undergoing transplantation following serous membrane detachment with an ultrasonic homogenizer, which mimics the Cavitron Ultrasonic Surgical Aspirator^®^ (CUSA), compared with no detachment. Furthermore, the bleeding amount was lower with the ultrasonic homogenizer method than with the needle and electric scalpel methods. These findings provide evidence that transplantation to the liver surface with serous membrane detachment using CUSA might contribute to the development of new treatments for cirrhosis using cells or tissues.

## 1. Introduction

Liver damage leads to the production of collagen fibers by activated stellate cells. During chronic liver damage, prolonged activation of stellate cells leads to the accumulation of collagen fibers produced by these cells in the liver, and the progression of ensuing fibrosis causes cirrhosis. The most common causes of cirrhosis are hepatitis B and C virus infections and alcoholic and nonalcoholic fatty liver disease [1]. Importantly, despite the decline in cirrhosis due to hepatitis B and C virus infections following the development of effective antiviral agents, the incidence rates of nonalcoholic fatty liver disease-related liver failure and hepatocellular carcinoma are increasing [2,3]. Nonalcoholic fatty liver disease is primarily caused by obesity and lifestyle-related diseases and, when left unresolved, advances to nonalcoholic steatohepatitis and ultimately cirrhosis and hepatocellular carcinoma [4].

Liver transplantation is the most effective treatment for end-stage liver cirrhosis; however, donor shortages, risk of complications, and necessity for lifelong immunosuppressive therapy remain major concerns [5]. Alternative approaches, including medical therapy using drugs simvastatin, rifaximin and so on [6], autologous stem cell transplantation [7,8], and hepatocyte transplantation [9], are not as effective as liver transplantation. Low cell engraftment and undesired differentiation are two main issues with stem cell transplantation [10], whereas transfer of the injected cells to other organs beyond liver via blood circulation has been reported in patients undergoing hepatocyte transplantation via injection into the spleen or portal vein [11,12]. Furthermore, patients with cirrhosis are at higher risk of complications such as portal hypertension and portal vein thrombosis and cell transplantation through the spleen and portal vein is risky. Therefore, our recent efforts have been focused on the development of new methods for cell transplantation to the liver surface.

By mimicking fetal liver development, we succeeded in producing a three-dimensional liver primordium from human induced pluripotent stem cell (hiPSC)-derived hepatic endoderm cells, vascular endothelial cells, and mesenchymal stem cells, which we termed hiPSC-derived liver buds (hiPSC-LB) [13,14]. Since the shape and gene expression profile of the hiPSC-LB were similar to those of mouse fetal liver [13], we used fetal liver as the transplant tissue in this study. However, we observed low engraftment rate and necrosis of the transplanted tissue following the transplantation of fetal liver to the liver surface; these issues could be related to the serous membrane covering the liver surface. Serous membrane, which comprises surface mesothelial cells and the underlying connective tissue, secretes serous fluid to protect organs from friction between intraperitoneal organs. In the present study, we evaluated the efficacy of serous membrane detachment before fetal liver transplantation on several outcomes after transplantation and compared several detachment methods including needle, electric scalpel used one or two times, tissue adhesive, and ultrasonic homogenizer. We evaluated needles as an alternative to the scalpel, which is used in surgery. We also evaluated electric scalpel used for incisions and hemostasis in surgery, and Vetbond^TM^, tissue adhesives proposed as an alternative to sutures for wound closure [15]. Finally, we evaluated the utility of ultrasonic homogenizer for the detachment of serous membrane covering the liver surface, based on the principle that CUSA is used to fracture and aspirate tissues by ultrasonic vibration during surgery [16].

First, we prepared a model of liver cirrhosis by administering N-nitrosodimethylamine (DMN) to DPPIV-deficient Fischer 344 rats [17]. DMN is a potent hepatotoxin, carcinogen, and mutagen [18] and causes acute and chronic liver damage with necrosis, fibrosis, and nodule regeneration [19]. DPPIV is a protease highly expressed in epithelial cells of many organs and is expressed in hepatocytes and cholangioepithelial cells in the liver [20,21]. We performed detachment of the serous membrane of the liver in DPPIV-deficient Fischer 344 rats using the abovementioned five methods and transplanted fetal livers harvested from 14-day-old embryos of DPPIV-positive pregnant Fischer 344 rats. We distinguished donor and recipient tissues by enzyme histochemistry for DPPIV [22,23].

We verified the utility of serous membrane detachment and compared the impact of different detachment methods on outcomes by evaluating survival and engraftment rates, engraftment area, liver/body weight and spleen/body weight ratios, liver function, and expression levels of several liver function markers. We found that serous membrane detachment using an ultrasonic homogenizer improved the engraftment rate of transplanted tissues and was a safe and clinically applicable detachment method. The findings of the present study provide important information for the realization of transplantation to the liver surface as a new transplantation method for liver cirrhosis.

## 2. Results

### 2.1. Effectiveness of Serous Membrane Detachment

We first tested our hypothesis that the serous membrane on the liver surface prevented the engraftment of transplanted fetal liver. We induced liver cirrhosis by intraperitoneal DMN administration in DPPIV-deficient rats and transplanted fetal livers from 14-day-old DPPIV-positive rat embryos onto the recipient livers with and without serous membrane detachment using a needle.

The livers were removed three weeks after transplantation to examine engraftment rate and engraftment area. Engraftment rate was calculated by macroscopic examination. Necrosis of the transplanted fetal livers was considered as engraftment failure even in the presence of engraftment signs. The engraftment rates of the transplanted fetal livers were 100% and 50% in rats with and without serous membrane detachment with the needle method, respectively (Figure 1b). DPPIV enzyme histochemistry to selectively stain the transplanted fetal livers confirmed that the engraftment area was significantly increased by serous membrane detachment (Figure 1c). Figure 1d shows representative macroscopic images of the cut surface of recipient livers three weeks after transplantation. To investigate whether serous membrane detachment altered the therapeutic effect of the transplanted fetal livers on cirrhosis, we measured liver function markers in blood samples that were collected weekly after transplantation. As shown in Figure 1e–h, serous membrane detachment with the needle method significantly reduced aspartate aminotransferase (AST) and total bilirubin (T-Bil) levels one week after transplantation, AST and alanine aminotransferase (ALT) levels two weeks after transplantation, and AST, ALT, and ammonia (NH_3_) levels three weeks after transplantation. These results suggested that detachment of serous membrane improved the efficiency of transplantation of fetal livers to the liver surface.

### 2.2. Amount of Bleeding during Surgery and Tissue Damage after Fetal Liver Transplantation Using Different Detachment Methods

We next measured the amount of bleeding during surgery and evaluated the tissue at the detachment site to compare the safety of five methods. Figure 2b shows the situation during detachment of serous membrane. After clamping the portal vein, serous membrane of the left lobe was detached by different detaching methods and bleeding was stopped by compression with a cotton swab for one minute. The weight of the cotton swab before hemostasis was subtracted from its weight after hemostasis to calculate bleeding amount per minute. Cotton swabs were replaced with a new one before it became saturated with blood. When detached the serous membrane with a needle and the electric scalpel ×2 (twice detached), the amount of bleeding was significantly higher than non-detachment (Figure 2c). Additionally, we performed hematoxylin/eosin staining (HE) on the liver samples harvested one week after the detachment of serous membrane of the left lobe to observe the detached site before healing. The needle and electric scalpel ×2 groups, which had a large amount of bleeding, had a clot remaining at the detachment site and exhibited more extensive inflammatory cell infiltration compared with the other detachment methods. In addition, tissue like granulation was observed on the liver with serous membrane detachment using the electric scalpel ×1 (once detached) (Figure 2d).

### 2.3. Engraftment Efficiency and Histological Assessment of the Fetal Liver Transplanted Using Different Detachment Methods

Based on our initial experiments showing the effectiveness of serous membrane detachment in fetal liver transplantation to the liver surface, we next determined the optimal detachment method in the same experiment protocol of Figure 1a. Given that detaching the serous membrane with a sharp instrument can cause major bleeding, an important concern in patients with cirrhosis who have an increased bleeding tendency, we aimed to develop a method that did not utilize a needle in a F344 rat model of liver cirrhosis. In recipient rats, serous membrane of the liver was detached using one of the following methods: electric scalpel used one or two times, tissue adhesive and ultrasonic homogenizer. The methods were compared with the needle group. After detachment, the fetal livers of 14-day-old embryos were transplanted to the recipient rats after hemostasis until the blood had stopped completely and engraftment rate was determined by macroscopic examination of the livers of recipient rats harvested three weeks after the transplantation. Hematoxylin/eosin staining (HE) showed that the transplanted tissue did not fall off and remained firmly at the transplantation site in recipients that underwent serous membrane detachment using the needle or ultrasonic homogenizer method (Figure 3a). In contrast, part of the transplanted tissue was observed to have fallen off in recipients that underwent serous membrane detachment using the electric scalpel or tissue adhesive methods and in those that did not undergo serous membrane detachment (Figure 3a). We found that the engraftment rate was highest in the needle group, followed by the ultrasonic homogenizer group (Figure 3b). We also determined the engraftment area by DPPIV enzyme histochemistry, which revealed that the engraftment area was largest in the needle group, followed by the ultrasonic homogenizer group (Figure 3c). These results suggested that ultrasonic homogenizer was the most suitable non-needle method for potential clinical use among those evaluated for serous membrane detachment.

### 2.4. Therapeutic Effect of the Transplanted Fetal Liver Using Different Detachment Methods

We next evaluated whether the therapeutic effect of the transplanted fetal liver depended on the serous membrane detachment method in a F344 rat model of liver cirrhosis. This experiment was performed in the same experiment protocol of Figure 1a. To that end, we first determined the rates of survival and weight gain at three weeks after transplantation. As shown in Figure 4a, the survival rates were 71.4% in the non-transplanted and adhesive groups, 83.3% in the non-detachment group, and 100% in the other groups. In addition, the weight gain rate was significantly higher in the needle, electric scalpel ×2, and ultrasonic homogenizer groups than in the non-transplanted group (Figure 4b). Furthermore, the liver/body weight ratio was significantly higher in the ultrasonic homogenizer group than in the non-transplanted group (Figure 4c) whereas the spleen/body weight ratio was significantly lower in all transplanted groups compared with the non-transplanted group (Figure 4d).

We also determined the plasma concentrations of AST, ALT, NH_3_, and T-Bil every week three weeks after transplantation. The AST and ALT levels were significantly lower in the needle, electric scalpel ×2, and ultrasonic homogenizer groups than in the non-transplanted group at three weeks after transplantation (Figure 4e,f). Additionally, the NH_3_ levels were significantly lower in the non-detachment, needle, twice-used electric scalpel, and ultrasonic homogenizer groups than in the non-transplanted group at one week after transplantation (Figure 4g). At three weeks after transplantation, the NH_3_ levels were significantly lower in the ultrasonic homogenizer group than in the non-transplanted group (Figure 4g). Additionally, the T-Bil levels at two weeks after transplantation were significantly lower in the once- and twice-used electric scalpel and ultrasonic homogenizer groups (Figure 4h). These results demonstrated the significant therapeutic effect of fetal liver transplantation following serous membrane detachment using the needle, twice-used electric scalpel, and ultrasonic homogenizer compared with the non-transplanted group. On the other hand, serous membrane detachment using tissue adhesive and once-used scalpel did not show good therapeutic benefit.

### 2.5. Evaluation for Maturation of the Transplanted Fetal Liver Using Different Detachment Methods

Finally, to evaluate the engrafted fetal liver tissues from DPPIV-positive rats in a Fischer 344 rat model of liver cirrhosis, we examined the maturity of the engrafted fetal livers by immunohistochemical staining for CD31, cytokeratin 19 (CK19), hepatocyte nuclear factor-4 alpha (HNF-4α), and albumin. This experiment was performed in the same experiment protocol of Figure 1a. We calculated the percentage of marker positive area in DPPIV (CD26) positive area which interpreted the engrafted fetal livers using the ImageJ imaging software. We found that CD26 was highly expressed in the needle and ultrasonic homogenizer groups and that the CD26-positive area was significantly larger in these two groups than in the non-detachment group (Figure 5a,b). On the other hand, CD31 was highly expressed in the non-detachment, electric scalpel ×1, and adhesive groups; the quantification by ImageJ also revealed that the CD31-positive area was significantly smaller in the ultrasonic homogenizer group than the non-detachment group (Figure 5a,c). CK19 was not expressed in any of the group (Figure 5a). While the expression level of HNF-4α was significantly higher in the ultrasonic homogenizer group than in other groups (Figure 5d,e), the expression level of albumin was significantly higher in the needle group, electric scalpel ×2, and ultrasonic homogenizer groups than in the non-detachment group (Figure 5d,f). These results suggested that the fetal liver transplanted after the detachment of serous membrane with an ultrasonic homogenizer developed into the most mature transplanted tissue at three weeks after transplantation.

## 3. Discussion

Liver transplantation is the most effective treatment for end-stage liver disease [24,25]. While several studies have investigated that injection of stem cells into the hepatic artery or spleen as an alternative treatment [26,27,28], few studies have evaluated transplantation to the liver surface. In the present study, we established a method to transplant fetal livers to the liver surface with high engraftment efficiency and low invasiveness as a new treatment approach for liver cirrhosis. As shown in Figure 1a,b, the engraftment rate and engraftment area of the fetal livers were significantly increased by detachment of serous membrane using a needle. In addition, the liver function markers were significantly improved using the needle method (Figure 1e–h). These results suggested that the engraftment efficiency and donor liver function could be improved with the detachment of serous membrane prior to transplantation. Liver is covered with a serous membrane composed of squamous epithelial cells, and the space between the serous membrane and hepatic parenchymal cells is supported by connective tissue, in which the portal vein and hepatic artery capillaries circulate [29]. We predict that the observed improvements in engraftment rate and area were due to the contact between the recipient blood vessels and the donor tissue as a result of the detachment of the serous membrane, which facilitated the supply of oxygen and nutrients to the donor.

Based on these initial findings, we aimed to develop a clinically applicable and efficient serous membrane detachment method as an alternative to the needle method. Although inferior to the needle group, the ultrasonic homogenizer group had the highest engraftment rate and the largest engraftment area among all detachment methods evaluated in the present study (Figure 3b,c). One reason for the highest engraftment efficiency observed with the needle method is that no heat is generated during the procedure. The low engraftment rate observed in the electric scalpel group provides evidence that tissue damage in the recipient due to high temperature might hinder the engraftment of donor tissue. Therefore, the engraftment efficiency of the ultrasonic homogenizer method for serous membrane detachment might be improved by cooling during the procedure. Although the tissue adhesive did not produce heat, the engraftment rate was not better than the needle or ultrasonic homogenizer methods. One potential explanation is the insufficient delivery of oxygen and nutrients to the fetal liver due to insufficient detachment of the serous membrane. The same possibility exists for the low engraftment rate observed in the non-detachment group. Detachment of serous membrane with a needle showed good engraftment in the donor; however, the amount of bleeding was high (Figure 2c). In addition, the damage to the detachment site was relatively severe based on the observation of inflammatory cells (Figure 2d). Therefore, detachment of serous membrane using a needle would be dangerous in patients with cirrhosis who are prone to bleeding. Serous membrane detachment using the electric scalpel two times led to significantly increased bleeding compared to detachment using the electric scalpel one time (Figure 2c). In addition, the evaluation of the detached site by staining revealed the presence of granulation tissue and increased inflammatory cell infiltration one week after detachment (Figure 2d). Detachment of the serous membrane with an electric scalpel may induce chronic inflammation in the recipient, resulting in poor donor engraftment. Electric scalpels have been reported to delay wound healing [30], and serous membrane detachment using an electric scalpel may not be suitable for patients with liver cirrhosis. In contrast, the ultrasonic homogenizer method was associated with the lowest amount of bleeding among the five methods evaluated in the present study and there was minimal damage at the site of detachment (Figure 2c,d). CUSA, which formed the basis of our ultrasonic homogenizer method, can be used in both cirrhotic and noncirrhotic livers and has a low blood loss rate with established safety [31]. Therefore, serous membrane detachment using CUSA may be considered as a safe approach even in patients with liver cirrhosis.

We previously reported that transplantation of rat fetal livers to the liver surface in a Fischer 344 rat model of liver cirrhosis improved survival rate and liver function [16]. As shown in Figure 4a,b, the rates of survival and weight gain after transplantation differed depending on the detachment method. The wound was shallow in rats that underwent detachment with a tissue adhesive; however, the tissue adhesive often spreads in unexpected directions and the detached surface became wider, which might be associated with the severe postoperative invasion of tissue and low survival rate. As shown in Figure 4e–h, the liver function markers were significantly improved in the needle, electric scalpel ×2, and ultrasonic homogenizer groups compared with the non-transplanted group, which might be related to depth of wound by detachment. As shown in Figure 2d, deep wounds were observed at the detachment site in the needle, electric scalpel ×2, and ultrasonic homogenizer groups. The fetal liver formed a large blood vessel of about 100 or 200 μm in diameter in recipient rats that underwent serous membrane detachment using the needle, twice-used electric scalpel, and ultrasonic homogenizer methods (Figure 5a). It is considered that these vessels supplied oxygen and nutrients to the engrafted fetal liver, which enabled fetal liver to exert therapeutic effect. On the other hand, the transplanted fetal liver formed capillary-like fine blood vessels (Figure 5a), the area of CD31 was high in the non-detachment, electric scalpel ×1, and adhesive groups (Figure 5c). From of the result that there were cavities in parts of the fetal liver (Figure 5a), it is considered that part of the transplanted fetal liver disappeared due to the lack of oxygen and nutrients as a result of insufficient detachment of the serous membrane in the non-detachment, electric scalpel ×1, and adhesive groups. We consider that capillary-like fine blood vessels were formed to supply oxygen and nutrients for oxygen-deficient fetal liver. The positive areas of CD26 and ALB were significantly higher in the needle and ultrasonic homogenizer groups compared with the non-detachment group (Figure 5b,f), the positive area of HNF-4α was significantly higher in only the ultrasonic homogenizer group (Figure 5e). It is considered that the maturity of the engrafted fetal liver might be related to depth of wound on the detached site and heat generated during detachment. We consider that the engrafted fetal liver matured due to some extent to the depth of wound on the detached site and no heat generated during detachment in the needle and ultrasonic homogenizer groups.

In conclusion, detachment of serous membrane with an ultrasonic homogenizer, similar to the use of a CUSA in clinical settings, improved engraftment of the transplanted fetal liver on the liver surface. Based on extensive information on its clinical safety and use in surgery, these results implicate serous membrane detachment using a CUSA as a safe approach in patients with cirrhosis undergoing transplantation. We previously reported that coating the donor rat fetal liver with ultrapurified alginate gel before transplantation to the liver surface of recipient rats improved both the engraftment area of the transplanted fetal liver and survival rate of the recipient rats [17]. We propose that transplantation to the liver surface should be possible by transplanting liver organoids coated with ultrapurified alginate gel after the detachment of serous membrane of the recipient liver surface with a CUSA. This novel transplantation method can potentially be used as a replacement for currently available liver transplantation approaches in patients with cirrhosis, addressing several issues including donor shortages as well as embolization of blood vessels [12] and transfer of transplanted cells to other organs. Fetal liver used in the present study is not used in clinical practice, and our findings should be confirmed using hepatocytes and organoids. This novel approach should also be tested for transplantation of organs other than liver. Future studies are warranted to confirm the safety of serous membrane detachment using an ultrasound homogenizer in models of severe cirrhosis with a strong bleeding tendency.

## 4. Materials and Methods

### 4.1. Experimental Animals

Three-week-old, female, DPPIV-deficient Fischer 344/DuCrlCrlj rats were purchased from Charles River Laboratories Japan, Inc. (Kanagawa, Japan). DPPIV-positive Fischer 344/NSlc rats were purchased from Japan SLC, Inc. (Shizuoka, Japan). All animal experiments were performed according to the ethical guidelines of the Animal Research Center in Medical College of Yokohama City University. The institutional animal care use committee of Yokohama City University approved all our animal studies (approval no. 17-025, 20-021).

### 4.2. Induction of Liver Cirrhosis in Rats

Before transplantation, 5-week-old DPPIV-deficient Fischer 344 rats were injected into the peritoneal cavity with 10 mg/kg DMN (FUJIFILM Wako Pure Chemical Corporation, Osaka, Japan). Three daily injections were followed by a four-day break and then another three daily injections were given. Transplantation was performed after three days of wash-out period. Starting the day after transplantation, the rats were injected with DMN in the peritoneal cavity for three days again and dissected three weeks after the transplantation.

### 4.3. Methods for Detachment of the Serous Membrane

Five detachment methods were used in the present study: needle, once-used electric scalpel, twice-used electric scalpel, adhesive, and ultrasonic homogenizer. In the needle method, 2–3 mm of serous membrane of the left lobe of the liver was detached using an 18G needle (Terumo, Tokyo, Japan) to achieve a depth of 1 mm. In the electric scalpel methods, an area with a diameter of 2–3 mm was cauterized on the surface of the left lobe of the liver using an electric scalpel (Braintree Scientific, Braintree, MA, USA) either once or twice, and the cauterized area of serous membrane was detached by tweezers. In the adhesive method, 2 µL Vetbond^TM^ (3M, Saint Paul, MN, USA) tissue adhesive was dropped on the surface of the left lobe of the liver and the solidified portion was detached by tweezers. In the ultrasonic homogenizer method, several drops of saline were dropped on the surface of the left lobe of the liver and 2–3 mm of serous membrane was detached using an ultrasonic homogenizer (Yamato Scientific, Tokyo, Japan) to achieve a depth of 1 mm. All detachment methods were performed on two separate places of the left lobe of the liver for each rat.

### 4.4. Transplantation of Rat Fetal Livers

Fetal liver tissues were harvested from 14-day-old embryos of pregnant DPPIV-positive Fischer 344 rats and used as donors for transplantation. To harvest the donor tissue, pregnant rats were euthanized, fetuses were collected, and whole fetal livers were isolated under a microscope. The collected whole fetal livers were stored in phosphate-buffered saline (PBS) on ice until transplantation. For transplantation, DPPIV-deficient Fischer 344 rats underwent laparotomy under anesthesia and portal blood flow was blocked with a vascular clamp. Following the detachment of serous membrane on the surface of left lobe of the liver by the indicated method, bleeding was stopped using pressure with cotton swabs and the whole fetal livers were transplanted. The left lobe was covered with the middle lobe, followed by removal of the vascular clamp and abdominal closure.

### 4.5. Measurement of Bleeding

The weight of cotton swabs were weighed before and after their use for hemostasis following serous membrane detachment, as described above. The cotton swabs were stored in tubes containing moistened tissue papers until measurement to prevent drying. The bleeding amount (mg/min) was determined by subtracting the weight after hemostasis from the weight before hemostasis.

### 4.6. Assessment of Liver Function

Concentrations of AST, ALT, NH_3_, and T-Bil in plasma were measured by a DRI-CHEM 7000V analyzer (FUJIFILM, Tokyo, Japan), as previously described [17].

### 4.7. Enzyme Histochemistry for DPPIV

Frozen tissue sections, 7 μm in thickness, were fixed in a mixed solution of acetone (FUJIFILM Wako Pure Chemical Corporation, Osaka, Japan): chloroform (FUJIFILM Wako Pure Chemical Corporation) at 1:1 ratio for 10 min at 4 °C. Next, the sections were air-dried and incubated in a staining solution prepared by mixing 8 mg glycyl-prolyl-4-methoxy-β-naphthylamide hydrochloride (Sigma-Aldrich, Saint Louis, MO, USA) dissolved in 1 mL dimethyl sulfoxide (FUJIFILM Wako Pure Chemical Corporation) and 1 mg Fast Blue BB Salt hemi zinc chloride salt (Sigma-Aldrich) dissolved in 1 mL PBS (pH = 7.0) at a 1:20 ratio. After incubation for 20 min at room temperature, the tissue sections were incubated for 5 min in 2% CuSO_4_ and rinsed with MQ and fixed with 10% formalin. After washing with MQ, the sections were counterstained with Carazzi’s hematoxylin (Muto Pure Chemicals, Tokyo, Japan). Images were acquired using a V120 Virtual Slide microscope (Olympus, Tokyo, Japan). Areas positive for DPPIV were quantified using ImageJ.

### 4.8. Immunohistochemical Staining

Frozen tissue sections were fixed in 1:1 acetone/methanol solution for 30 min at −30 °C. After washing with PBS with 0.05% Tween20, the slides were blocked with the serum-free Protein Block reagent (Dako) for 1 h at room temperature. Next, the slides were incubated overnight at 4 °C with primary antibodies diluted in the blocking reagent. The following antibodies were used: anti-CD26 (559639, 1:100; BD Biosciences, San Jose, CA, USA), anti-CD31 (550300, 1:50; BD Biosciences), anti-CK19 (61029, 1:200; Progen Biotechnik, Heidelberg, Germany), anti-albumin (NBP1-32458, 1:200; Novus Biologicals, Littleton, CO, USA), anti-HNF-4α (clone H-1, sc-374229, 1:200; Santa Cruz, CA, USA). Next, the slides were incubated for 1 h at room temperature with appropriate Alexa Fluor 488-, 555-, or 647-conjugated secondary antibodies and counterstained with DAPI (4′,6-diamidino-2-phenylindole dihydrochloride, 1:1000; FUJIFILM Wako Pure Chemical Corporation). Images were acquired using the V120 Virtual Slide microscope. Areas positive for CD26, CD31, HNF-4α, or albumin were divided by the entire area of transplanted fetal liver to calculate areas positive for the indicated marker.

### 4.9. Statistical Analysis

Statistical analyses were performed by non-repeated measures analysis of variance with Bonferroni correction. Data were presented as means ± standard error of the mean. *p* values < 0.05 were considered to indicate statistical significance.

## Figures and Tables

**Figure 1 ijms-22-11589-f001:**
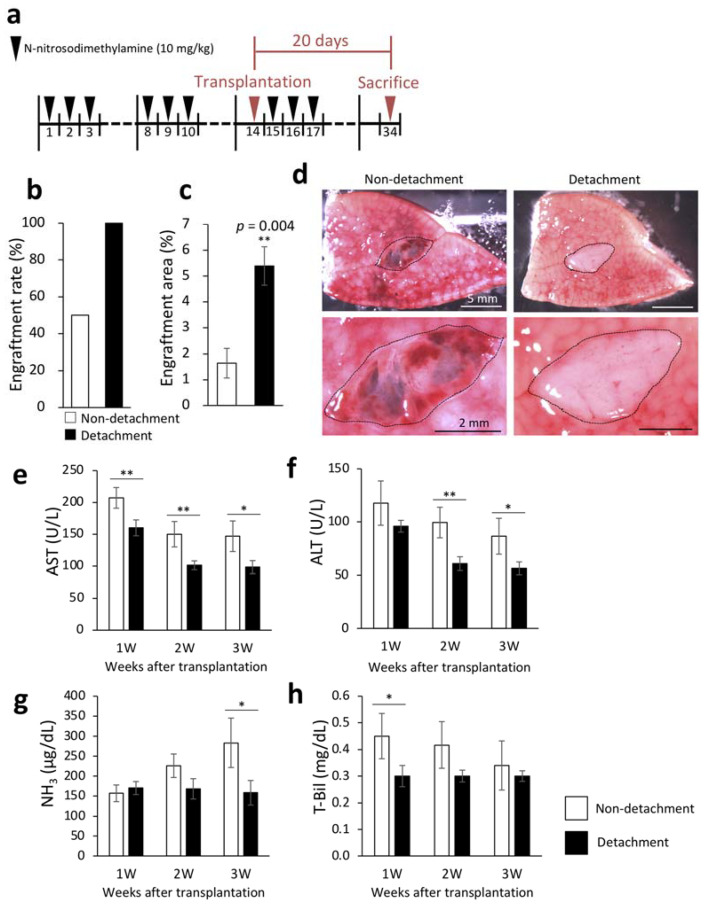
Engraftment efficiency of the fetal liver and evaluation of liver function following serous membrane detachment in the DMN-induced liver cirrhosis rat model. (**a**) The procedure for producing rat model of liver cirrhosis. Livers were collected 20 days after transplantation. (**b**) At three weeks after transplantation, fetal liver engraftment rate was determined in the non-detachment (*n* = 6) and detachment (*n* = 7) groups. (**c**) Livers were removed three weeks after transplantation, and frozen sections were stained for dipeptidyl peptidase IV (DPPIV). Areas with DPPIV-positive staining were quantified using ImageJ (non-detachment, *n* = 6; detachment, *n* = 7). *p* = 0.004. (**d**) Fetal livers were transplanted between the left and middle lobes of rats, cut surface of the liver was observed three weeks after transplantation by microscopy. The area encircled by the black dotted line indicates the fetal liver. (**d**–**g**) Plasma concentrations of aspartate aminotransferase (AST) (**e**), alanine aminotransferase (ALT) (**f**), NH_3_ (**g**), and total bilirubin (T-Bil) (**h**) in recipient rats were measured at 1–3 weeks after transplantation (non-detachment, *n* = 6; detachment, *n* = 7). The data are shown as the mean ± standard error. * *p* < 0.05, ** *p* < 0.01 versus non-detachment group by non-repeated measures analysis of variance with Bonferroni correction.

**Figure 2 ijms-22-11589-f002:**
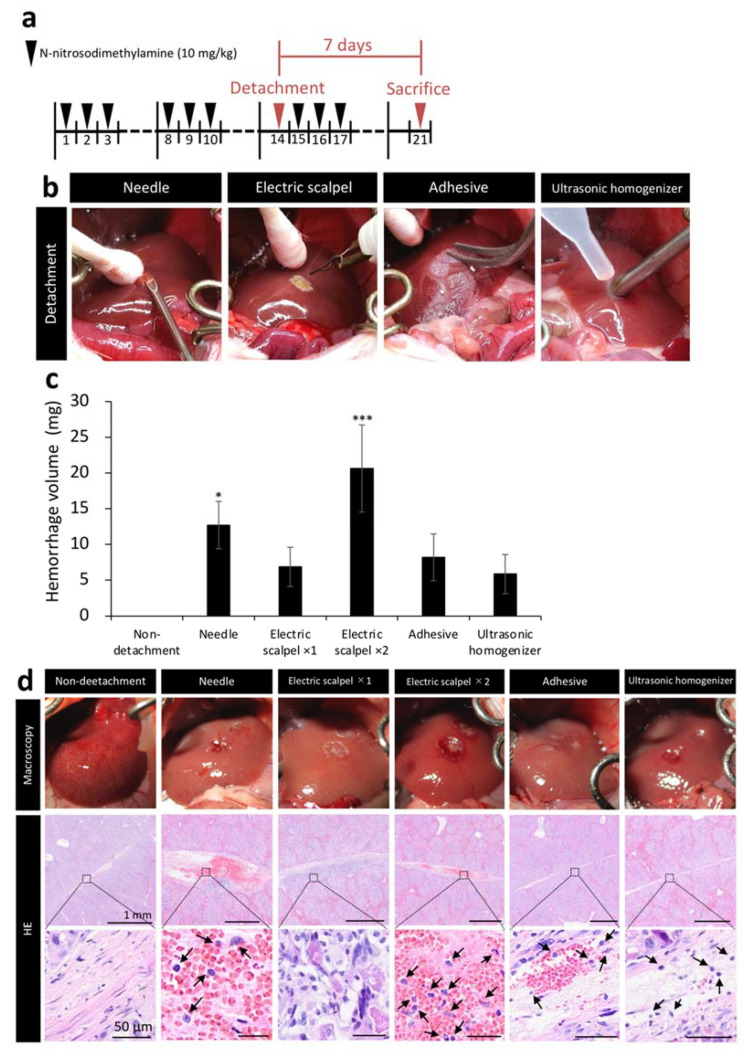
Assessment of tissue damage in five detachment methods. (**a**) Rat model of liver cirrhosis was produced as described in Figure 1. Livers were collected 7 days after transplantation in Figure 2. (**b**) Image showing detachment of serous membrane using the needle, electric scalpel, tissue adhesive, and ultrasonic homogenizer methods. (**c**) Bleeding amount per minute (*n* = 3). (**d**) Immediately after the detachment of serous membrane of the left lobe, the livers were observed macroscopically. In addition, the livers were removed one week after the serous membrane detachment to stain the detached site with hematoxylin/eosin (HE). Arrows indicate inflammatory cells. The data are shown as the mean ± Standard error. * *p* < 0.05, *** *p* < 0.001 versus non-detachment group by non-repeated measures analysis of variance with Bonferroni correction.

**Figure 3 ijms-22-11589-f003:**
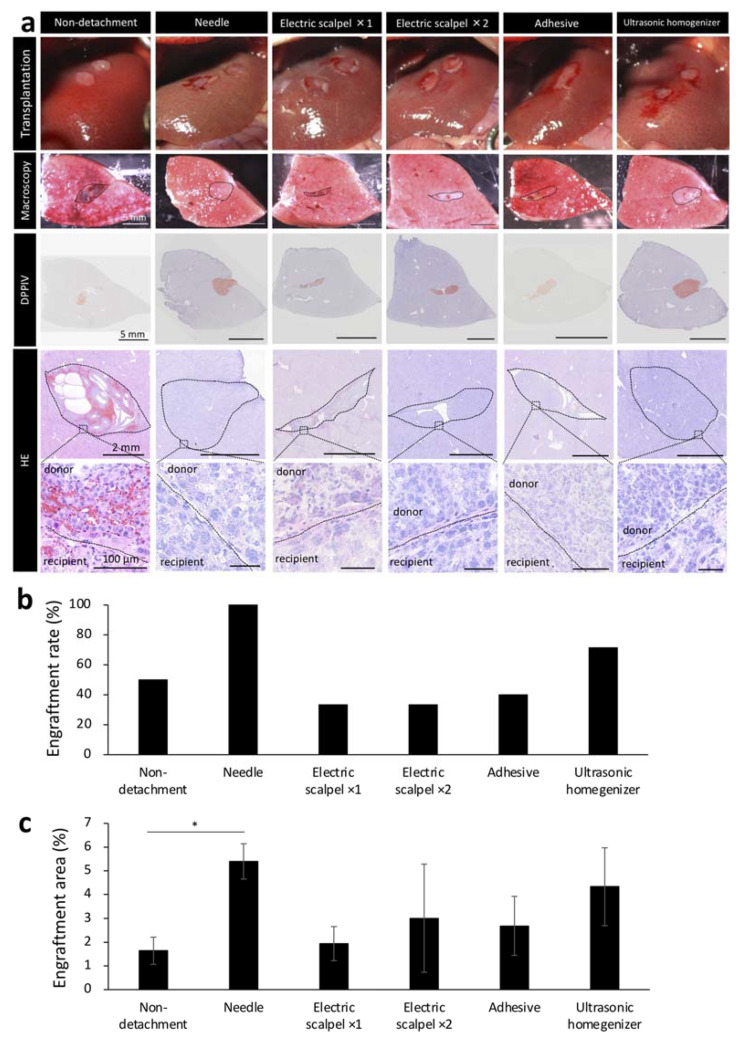
Engraftment efficiency and histological evaluation of fetal livers transplanted using different detachment methods. (**a**) The first images show the condition after transplantation of fetal livers with or without serous membrane detachment. The second images show macroscopic images of the cut surface of the liver three weeks after transplantation. Frozen sections of these tissues were stained with hematoxylin/eosin (HE) and Dipeptidyl Peptidase-4 (DPPIV). DPPIV-positive areas are stained orange. (**b**) Fetal liver engraftment rates at three weeks after transplantation (non-detachment, *n* = 6; needle, *n* = 7; electric scalpel ×1, *n* = 6; electric scalpel ×2, *n* = 6; adhesive, *n* = 6; ultrasonic homogenizer, *n* = 7). (**c**) Areas positive for DPPIV was divided by the area of entire recipient liver including the donor to calculate DPPIV expression level. ImageJ was used for analysis (non-detachment, *n* = 6; needle, *n* = 7; electric scalpel ×1, *n* = 6; electric scalpel ×2, *n* = 6; adhesive, *n* = 6; ultrasonic homogenizer, *n* = 7). The data are shown as the mean ± standard error. * *p* < 0.05 versus non-detachment group by non-repeated measures analysis of variance with Bonferroni correction.

**Figure 4 ijms-22-11589-f004:**
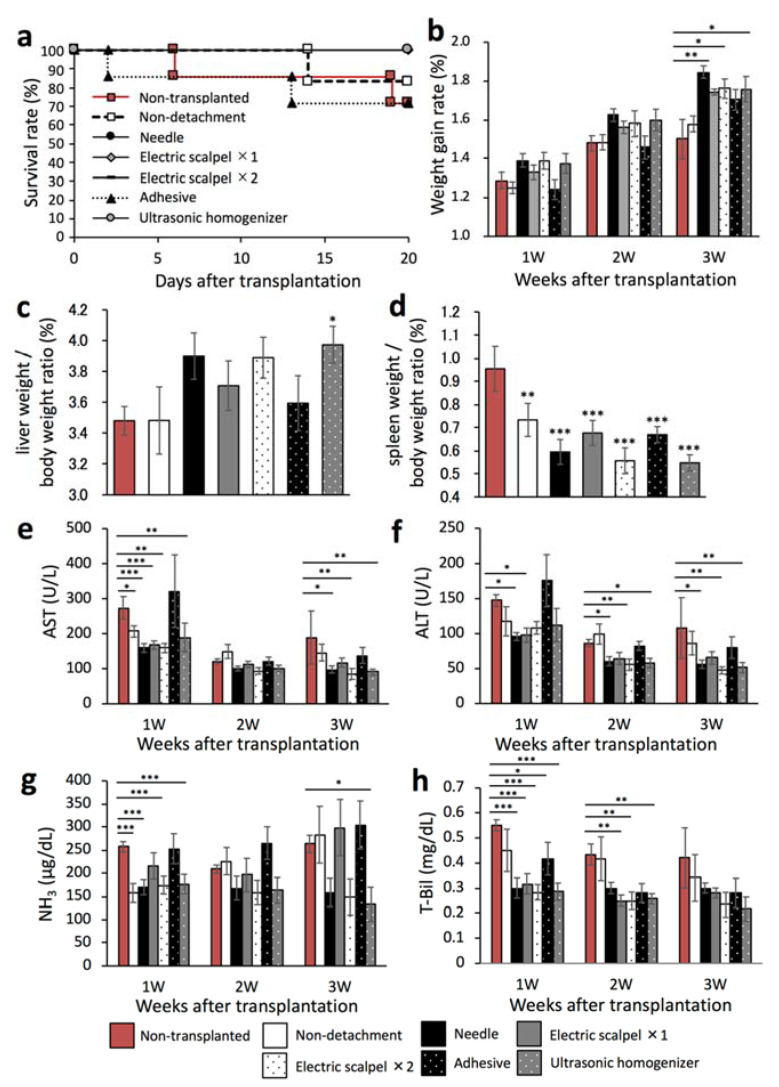
Evaluation of the therapeutic effect of fetal livers transplanted using different detachment methods. (**a**) Survival rates of the rats after transplantation (non-transplanted, *n* = 6; non-detachment, *n* = 6; needle, *n* = 7; electric scalpel ×1, *n* = 6; electric scalpel ×2, *n* = 6; adhesive, *n* = 7; ultrasonic homogenizer, *n* = 7). (**b**) Rate of weight gain after transplantation (non-transplanted, *n* = 7; non-detachment, *n* = 6; needle, *n* = 7; electric scalpel ×1, *n* = 6; electric scalpel ×2, *n* = 6; adhesive, *n* = 7; ultrasonic homogenizer, *n* = 7). (**c**,**d**) Three weeks after transplantation, liver and spleen were removed and weighed to determine liver/body weight ratio and spleen/body weight ratio (non-transplanted, *n* = 7, non-detachment, *n* = 6, needle, *n* = 7, electric scalpel ×1, *n* = 6, electric scalpel ×2, *n* = 6, adhesive, *n* = 7, ultrasonic homogenizer, *n* = 7). (**e**–**h**) At 1, 2, and 3 weeks after transplantation, plasma AST (**e**), ALT (**f**), NH_3_ (**g**) and T-Bil concentrations (**h**) were measured (non-transplanted, *n* = 7; non-detachment, *n* = 6; needle, *n* = 7; electric scalpel ×1, *n* = 6; electric scalpel ×2, *n* = 6; adhesive, *n* = 7, ultrasonic homogenizer, *n* = 7). The data are shown as the mean ± Standard error. * *p* < 0.05, ** *p* < 0.01, *** *p* < 0.001 versus non-detachment group by non-repeated measures analysis of variance with Bonferroni correction.

**Figure 5 ijms-22-11589-f005:**
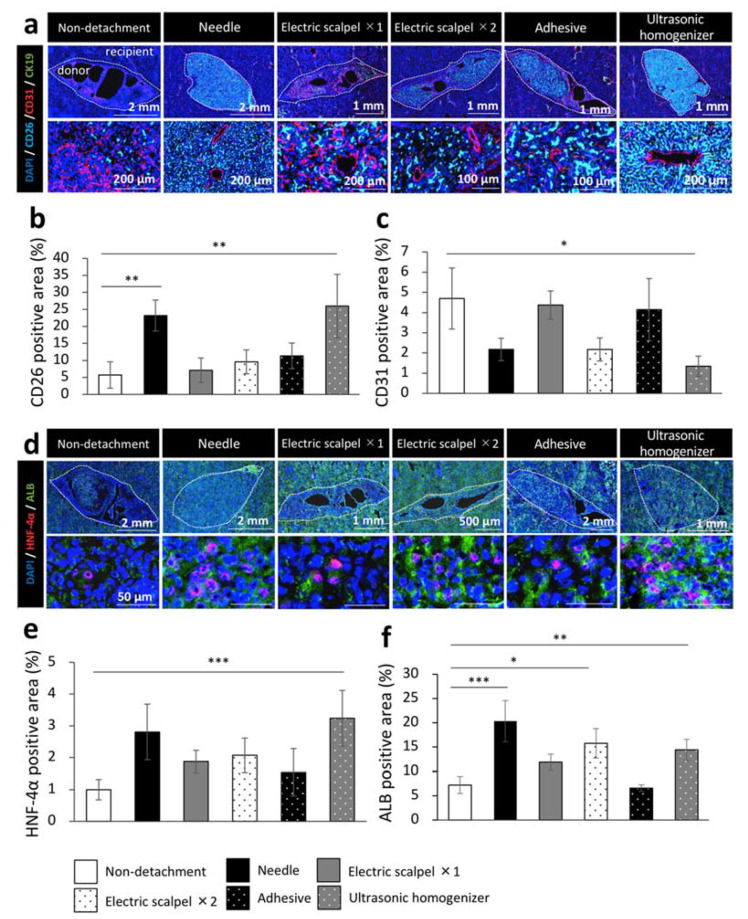
Immunohistochemistry of the fetal livers transplanted using different detachment methods. (**a**,**d**) Immunohistochemical staining for DPPIV (CD26, light blue), vascular endothelial cell marker (CD31, red), bile duct epithelial cell marker (CK19, green), hepatocyte nucleus marker (HNF-4α, red) and hepatocyte marker (albumin, green). (**b**,**c**,**e**,**f**) Areas positive for CD26, CD31, HNF-4α, or albumin of the transplanted tissues were measured at three weeks after transplantation (non-detachment, *n* = 6; needle, *n* = 4; electric scalpel ×1, *n* = 6; electric scalpel ×2, *n* = 4; adhesive, *n* = 4; ultrasonic homogenizer, *n* = 4). The data are shown as the mean ± standard error. * *p* < 0.05, ** *p* < 0.01, *** *p* < 0.001 versus non-detachment group by non-repeated measures analysis of variance with Bonferroni correction.

## Data Availability

The data presented in this study are available on request from the corresponding author.

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
