# Peer review of "Serous Membrane Detachment with Ultrasonic Homogenizer Improves Engraftment of Fetal Liver to Liver Surface in a Rat Model of Cirrhosis"

_ijms, 2021, doi:10.3390/ijms222111589_

Round 1
Reviewer 1 Report
The present manuscript compares the different results obtained by performing fetal liver transplantation in cirrhotic rats by detaching the serous membrane of the liver from the recipients with different surgical techniques.
In my view, this is an interesting and new work. Here are my comments on the subject.
The title seems to me to be correct and to reflect the content of what is to come later in the paper.
The abstract is adequate, although to speak at the end of organoids, without any other indication or introduction of what they are, may confuse the reader.
Line 44: please specify what you mean by medical therapy.
Figure 1 A: why no error bar?
Weighing the cotton swab to determine the amount of bleeding: Has this method been used in other articles? Isn't there a possibility that the cotton swab could become saturated and fail to account for blood loss?
Line 139 and 143: Explain what x2 and x1 mean.
The histology image in Figure 2 should be larger. At this size it is not very well appreciated.
Define SE or avoid using the abbreviation as you have done in the statistics section.
Figure 2: define HE
This paragraph is not very clear:
"Given that detaching the serous membrane with a sharp instrument can cause major bleeding, an important concern in patients with cirrhosis who have an increased bleeding tendency, we aimed to develop a method that did not utilize a needle in a F344 rat model of liver cirrhosis. In recipient rats, serous membrane of the liver was detached using one of the following methods: needle, electric scalpel used one or two times, tissue adhesive and ultrasonic homogenizer. "
This sentence may be confusing;
"We also determined the engraftment area by DPPIV 175 enzyme histochemistry, which revealed that the engraftment area was largest in the needle group, followed by the ultrasonic homogenizer group (Fig. 3c). These results suggested that ultrasonic homogenizer was the most effective method for potential clinical use among those evaluated for serous membrane detachment."
Figure 3: Larger histology images. Error bar in figure 3B is missing. Define abbreviations
Figure 5: Increase the size of histology images.
Section 4.6: enlarge and indicate any references where the described methodology has been previously used.
Line 429: delete “unfixed”
Author Response
- The abstract is adequate, although to speak at the end of organoids, without any other indication or introduction of what they are, may confuse the reader.
→Thank you for the comment. I revised to cells and tissues instead of liver organoids which are more recognized as transplant donors than liver organoids(page 1, line 25-26).
- Line 44: please specify what you mean by medical therapy.
→Thank you for your advice. I explained medical therapy in detail by describing the name of the drug used for liver cirrhosis drug therapy (page 1, line 44 to page 2 line 45).
- Figure 1 A: why no error bar?
→Thank you for your comment. Figure 1 A has no error bars because it is calculated by dividing the number of engrafted individuals by the total number of individuals. This is similar to survival rate graphs which also do not have error bars (page 4, line 121-122).
- Weighing the cotton swab to determine the amount of bleeding: Has this method been used in other articles? Isn't there a possibility that the cotton swab could become saturated and fail to account for blood loss?
→Thank you for your comment. This method has not been stated in other articles to our knowledge. We replaced a swab with a new one before it became saturated with the blood. Hemostasis was performed using multiple cotton swabs. Cotton swabs were replaced with a new one before it became saturated with blood (page 5, line 138-140).
- Line 139 and 143: Explain what x2 and x1 mean.
→Thank you for your comment. x2 means that the serous membrane was detached twice. x1 means that the serous membrane was peeled only once. We added “twice detached” and “once detached” to make it more clear (page 5, line 142).
- The histology image in Figure 2 should be larger. At this size it is not very
well appreciated.
→Thank you for the comment. The histology images were enlarged in Figure 2 (page 6).
- Define SE or avoid using the abbreviation as you have done in the statistics
section.
→Thank you we changed SE to standard error (Line130, 159, 198, 240, 268).
- Figure 2: define HE
→Thank you for the comment. We changed HE to hematoxylin/eosin(HE). (line 158, 174, 192)
- This paragraph is not very clear:
"Given that detaching the serous membrane with a sharp instrument can cause major bleeding, an important concern in patients with cirrhosis who have an increased bleeding tendency, we aimed to develop a method that did not utilize a needle in a F344 rat model of liver cirrhosis. In recipient rats, serous membrane of the liver was detached using one of the following methods: needle, electric scalpel used one or two times, tissue adhesive and ultrasonic homogenizer."
→Thank you for the valuable comment. I modified this paragraph to become clear.
Given that detaching the serous membrane with a sharp instrument can cause major bleeding, an important concern in patients with cirrhosis who have an increased bleeding tendency, we aimed to develop a method that did not utilize a needle in a F344 rat model of liver cirrhosis. In recipient rats, serous membrane of the liver was detached using one of the following methods: electric scalpel used one or two times, tissue adhesive and ultrasonic homogenizer. The methods were compared with the needle group.
The way “needle” was mentioned in the group of new methods may have made the paragraph confusing. Therefore I stated the needle group separately after that sentence. (Page 7, line165-171)
- This sentence may be confusing;
"We also determined the engraftment area by DPPIV 175 enzyme histochemistry, which revealed that the engraftment area was largest in the needle group, followed by the ultrasonic homogenizer group (Fig. 3c). These results suggested that ultrasonic homogenizer was the most effective method for potential clinical use among those evaluated for serous membrane detachment."
→Thank you for the valuable comments. We modified this sentence to become clear.
We also determined the engraftment area by DPPIV 175 enzyme histochemistry, which revealed that the engraftment area was largest in the needle group, followed by the ultrasonic homogenizer group (Fig. 3c). These results suggested that ultrasonic homogenizer was the most suitable non-needle method for potential clinical use among those evaluated for serous membrane detachment.
We added the phrase “non-needle method” to avoid confusion. (Page 7, line181-186).
- Figure 3: Larger histology images. Error bar in figure 3B is missing. Define abbreviations
→Thank you for the comment.
・We enlarged these histology images (Figure 3a, page 8).
・Figure 3B has no error bars because it is calculated by dividing the number of engrafted individuals by the total number of individuals.
・We defined the abbreviation to Dipeptidyl Peptidas-4 (DPPIV) (page 8, line 192).
- Figure 5: Increase the size of histology images.
→Thank you for the comment. The histology images in Fig.5d were enlarged (page 11). As we would like to show the engrafted fetal livers formed a large blood vessel, the images in Fig.5a were not enlarged.
- Section 4.6: enlarge and indicate any references where the described methodology has been previously used.
→Thank you for the comment.
We modified this sentence to “Concentrations of AST, ALT, NH3, and T-Bil in plasma were measured by a DRI-CHEM 7000V analyzer (FUJIFILM, Tokyo, Japan), as previously described[17]”.(page 15, line 418). The DRI-CHEM 7000V analyzer was used for our research.
- Line 429: delete “unfixed”
→We deleted “unfixed” (page 15, line 434).

Reviewer 2 Report
The authors investigated the effect of detachment of serous membrane on engraftment of fetal liver. The article can be published after correction for the following issues:
- Experimental design is not clearly described:
- a) How many experiments were performed? It seems to me that the results of two separate experiments are described; the results of the first experiment are presented in Fig 1 and of the second experiment in the figures 2-5. If this is true it is not clear why the N differs. (N=3 in Fig 2, N=6-7 in Fig3-4, and 4-6 in Fig 5.) The experiment seems the same and it is not clear why some samples were not analysed in Figures 2 and 5). If experiment in Fig 2 is a separate experiment it is not clear why the blood loss was not also measured in the next experiment (Fig3-5); please see also the concern in point 2b. If the results of different experiments are presented then the detailed description of conducted experiments and what was analysed should be provided.
- The DPTV protocol is not clearly described: “for three consecutive days for two weeks“. This description is confusing. I understood that three daily injections were followed by four days break and then another three daily injections were given. Is this correct? If yes please describe so, if not describe more clearly .
- Presentation and interpretation of data:
- a) Fig 1: there is an increase in NH3 in the non detachment group. How do you comment this?
- b) Fig 2b) It is quite obvious that all of the detachment groups are significantly different when compared to the non-detachment group where the amount of bleeding is zero. The only possible reason for obtaining insignificant p-values here would be the lack of power due to low number of samples. New samples should be added and interpretation adjusted.
- c) Fig 5 were the ALB and HNF-4alpha positive cells counted in total liver or only in the DPPIV positive (engrafted cells). Please provide this information and explain the rationale.
- d) Ultrasonic detachment does not appear to be better than needle detachment for majority of parameters and for some is worse (Fig 3). The methods were not directly compared, but only indirectly through their relationship against the non-detachment group. The differences seem to be marginal and I would suggest interpreting them more carefully.
Author Response
- Experimental design is not clearly described
→Thank you for the valuable comment.
We performed experiment follow the flow below.
- Investigate the effectivity of the detachment of the serous membrane.
- Investigation of surgical damage and tissue damage due to different detachment methods.
- Compare the engraftment of fetal livers in different detachment methods.
- Compare the therapeutic effect after the transplantation in different detachment methods.
We added these sentence to describe experimental design clearly.
・Page 4, Line119-120
(a) The procedure for producing rat model of liver cirrhosis. Livers were collected 20 days after transplantation.
・Page 6, Line151-152
(a) Rat model of liver cirrhosis was produced as described in Figure.1. Livers were collected 7 days after transplantation in Figure 2.
Experimental design was showed in Fig1a and Fig.2a.
- a) How many experiments were performed? It seems to me that the results of two separate experiments are described; the results of the first experiment are presented in Fig 1 and of the second experiment in the figures 2-5. If this is true it is not clear why the N differs. (N=3 in Fig 2, N=6-7 in Fig3-4, and 4-6 in Fig 5.) The experiment seems the same and it is not clear why some samples were not analyzed in Figures 2 and 5).
If experiment in Fig 2 is a separate experiment it is not clear why the blood loss was not also measured in the next experiment (Fig3-5); please see also the concern in point 2b. If the results of different experiments are presented then the detailed description of conducted experiments and what was analyzed should be provided.
→Thank you for the comments.
Experimental designs were showed in Fig1a and Fig.2a, it was shown that Fig.2 and Fig.3-5 are different protocols.
Experiment in Fig.2 was performed separately from experiment in Fig.3-5. In Fig.2, rats were dissected one week after the surgery to observe the detached site before healing, while in Fig.3-5, rats were dissected three weeks after to observe the engraftment rate and liver function.
We added the explanation.
・Page 5, Line145
Additionally, we performed hematoxylin/eosin staining (HE) on the liver samples harvested one week after the detachment of serous membrane of the left lobe to observe the detached site before healing.
・page 7, Line172
After detachment, the fetal livers of 14-day-old embryos were transplanted to the recipient rats after hemostasis until the blood has stopped completely. and engraftment rate was determined by macroscopic examination of the livers of recipient rats harvested three weeks after the transplantation.
- The DPTV protocol is not clearly described: “for three consecutive days for two weeks“. This description is confusing. I understood that three daily injections were followed by four days break and then another three daily injections were given. Is this correct? If yes please describe so, if not describe more clearly.
→Thank you for the important comment. You are completely right. Figure which shows schedule of injection was added in Fig.1 a.
Before transplantation, 5-week-old DPPIV-deficient Fischer 344 rats were injected into peritoneal cavity with 10 mg/kg DMN (FUJIFILM Wako Pure Chemical Corporation, Osaka, Japan). Three daily injections were followed by four days break and then another three daily injections were given. (Page 14, Line382-383).
- a) Fig 1: there is an increase in NH3 in the non-detachment group. How do you comment this?
→thank you for the comment.
The fetal liver engraftment rate is lower in the non-detachment group than in the detachment group, and even if engrafted, the tissue condition is poor. Even in the non-detachment group, the fetal liver contributed to the improvement of the liver function of the recipient 1 week after the transplantation, but the condition of the engrafted fetal liver gradually deteriorated and the effect of improving the liver function was lost. So 2 and 3 weeks after the transplantation, the NH3 level may have risen in the non-detachment group.
- b) Fig 2b) It is quite obvious that all of the detachment groups are significantly different when compared to the non-detachment group where the amount of bleeding is zero. The only possible reason for obtaining insignificant p-values here would be the lack of power due to low number of samples. New samples should be added and interpretation adjusted.
→Thank you for the valuable comment.
What can be said in this experiment is that bleeding tendency is seen when the liver surface is peeled off with Needle or Electric scalpel, so I'm sorry, but the number of experiments did not increase.
- c) Fig 5 were the ALB and HNF-4alpha positive cells counted in total liver or only in the DPPIV positive (engrafted cells). Please provide this information and explain the rationale.
→Thank you for the comments.
Since the transplanted tissue is DPPIV-positive, HNF4 and albumin-positive cells among the DPPIV-positive cells were measured to examine the maturity of the transplanted tissue.
We modified these sentences as below.
Finally, to evaluate the engrafted fetal liver tissues from DPPIV-positive rats in a Fischer 344 rat model of liver cirrhosis, we examined the maturity of the engrafted fetal livers by immunohistochemical staining for CD31, cytokeratin 19 (CK19), hepatocyte nuclear factor-4 alpha (HNF-4α), and albumin. We calculated the percentage of marker positive area in DPPIV (CD26) positive area which interpreted the engrafted fetal livers using the ImageJ imaging software. (Line 247 to 249).
- d) Ultrasonic detachment does not appear to be better than needle detachment for majority of parameters and for some is worse (Fig 3). The methods were not directly compared, but only indirectly through their relationship against the non-detachment group. The differences seem to be marginal and I would suggest interpreting them more carefully.
Needle detachment does not cause heat damage to the tissue, and the engraftment of the transplanted tissue is excellent, but the drawback is that there is a large amount of bleeding associated with the detachment. Sharp detachment, such as needle, can cause serious complications, especially in the bleeding-prone disease of cirrhosis.Ultrasound detachment is similar to a surgical instrument called CUSA and is often used in liver surgery with cirrhosis. The usefulness of ultrasonic detachment is that there is less heat damage to the tissue and less bleeding. In terms of the above, the ultrasonic detachment is considered to be an superior method to the other peeling methods compared and examined this time.
